# Glioblastoma in the Elderly: Review of Molecular and Therapeutic Aspects

**DOI:** 10.3390/biomedicines10030644

**Published:** 2022-03-10

**Authors:** Francesco Bruno, Alessia Pellerino, Rosa Palmiero, Luca Bertero, Cristina Mantovani, Diego Garbossa, Riccardo Soffietti, Roberta Rudà

**Affiliations:** 1Division of Neuro-Oncology, Department of Neuroscience, University and City of Health and Science, 10126 Turin, Italy; alessia.pellerino@unito.it (A.P.); rosapalmiero61@gmail.com (R.P.); riccardo.soffietti@unito.it (R.S.); rudarob@hotmail.com (R.R.); 2Pathology Unit, Department of Medical Sciences, University of Turin, 10126 Turin, Italy; luca.bertero@live.it; 3Division of Radiotherapy, Department of Oncology, University and City of Health and Science, 10126 Turin, Italy; cristina.mantovani@ymail.it; 4Division of Neurosurgery, Department of Neuroscience, University and City of Health and Science, 10126 Turin, Italy; diego.garbossa@unito.it; 5Department of Neurology, Castelfranco and Treviso Hospitals, 31100 Treviso, Italy

**Keywords:** glioblastoma, elderly patients, molecular factors, prognostic factors, comprehensive geriatric assessment

## Abstract

Glioblastoma (GBM) is the most aggressive primary brain tumour. As GBM incidence is associated with age, elderly people represent a consistent subgroup of patients. Elderly people with GBM show dismal prognosis (about 6 months) and limited response to treatments. Age is a negative prognostic factor, which correlates with clinical frailty, poorer tolerability to surgery or adjuvant radio-chemotherapy, and higher occurrence of comorbidities and/or secondary complications. The aim of this paper is to review the clinical and molecular characteristics, current therapeutic options, and prognostic factors of elderly patients with GBM.

## 1. Introduction

Glioblastoma (GBM) is the most frequent and aggressive primary brain tumour in adults, accounting for 14.5% of all brain tumours and 57.7% of gliomas [1]. 

Elderly people represent a consistent population of GBM patients. In fact, according to the Central Brain Tumour Registry of the United States (CBTRUS) statistical report of the 2013–2017 period, the incidence of GBM is 3.23 per 100,000 people per year, being higher among people older than 40 years (6.97 per 100,000 people per year) and reaching its peak in people 75–84 years old (15.30 per 100,000 people per year) [1]. Moreover, as life expectancy is progressively increasing in most countries around the world, the incidence of GBM in the elderly population will further rise in the coming years.

Elderly people with GBM show dismal prognosis (about 6 months) and limited response to treatments. The reasons are many, and not entirely known. First, age is an independent negative prognostic factor, probably due to more aggressive tumour biology in the elderly population. Second, frail elderly patients with poor clinical status may not tolerate surgery, or adjuvant radio-chemotherapy, which makes physicians more reluctant to offer aggressive treatments. Third, the occurrence of comorbidities and/or secondary complications may jeopardise surgery and radio-chemotherapy, thus determining early interruption of therapies. 

The aim of this paper is to review the clinical and molecular characteristics, current therapeutic options, and prognostic factors of elderly patients with GBM. 

## 2. Molecular Characteristics 

Tumour biology changes with age. Elderly patients with GBM harbour specific molecular features that correlate with a worse outcome.

In a sub-analysis of 272 GBMs from The Cancer Genome Atlas (TCGA), Noushmehr et al. found that tumours of older patients had a significantly lower rate of the glioma-CpG island methylator phenotype (G-CIMP). G-CIMP is commonly seen in isocitrate dehydrogenase (*IDH*)-mutant lower-grade gliomas, whereas it is a rarer find in GBMs. In addition, G-CIMP may correlate with the O-6-methylguanine-DNA methyltransferase promoter (*MGMT*p) methylation, which is the strongest predictive factor for response to chemotherapy with alkylating agents, both in newly-diagnosed and relapsing GBMs [2,3,4]. Interestingly, in the GBM subgroup described by Noushmehr et al., tumours presenting G-CIMP prevailed among younger patients (median age being 36 versus 56 years, *p* < 0.001) and correlated with the presence of the *IDH1* mutation. Furthermore, GBMs with G-CIMP often lacked some molecular alterations usually found in typical GBM (such as epidermal growth factor receptor (*EGFR*) amplification, chromosome 7 gain, chromosome 10 loss). Survival of patients with G-CIMP was significantly longer than that of older patients without the G-CIMP phenotype. However, G-CIMP was not the only factor associated with better survival, as age turned out to be a significant independent prognostic factor for survival regardless of the G-CIMP status of the tumour [5]. 

Since then, other studies have comprehensively explored the role of genetic and molecular factors to find a potential association with age in GBM patients. 

Bozdag et al. analysed a large cohort of 425 GBM patients from TCGA [6]. Tumours with the G-CIMP phenotype were excluded to avoid a potential confounding positive effect on prognosis. Then, patients were divided into two different categories according to age (≤40 years, ‘young’, and ≥70 years, ‘old’). First, the authors confirmed that age was a strong independent prognostic factor, even within G-CIMP-negative GBMs (*p* < 0.001). Second, they examined the prevalence of differentially expressed genes (DEGs) in younger and older patients with GBM. They found that some genes were upregulated in older patients (*PRUNE2*, *TMEM144*, *SLC14A1*), while other were downregulated (*H2AFY2*, *ENOSF1*, *SFRP1*, *RANBP17*, *SVIL*, *TUSC3*, *ATF7IP2*, *FZD6*, *TSPYL5*, *DLK1*, *HIST3H2A*). Interestingly, some genes were already known to have a biological role in tumourigenesis, either as tumour suppressors (such as *TUSC3*) or elements of critical importance within tumour growth pathways (i.e., *SFRP1* and *FZD6*, belonging to the Wnt signalling pathway) [7,8]. Third, age-specific microRNA expression, differentially-methylated genes (DMGs), and differentially-altered genes (DAGs) were analysed by the authors. In particular, they found that 19 microRNAs were downregulated in older GBMs (ebv-miR-BART1-5p, hsa-miR-422b, hsa-miR-507, hsa-miR-147, ebv-miR-BHRF1-2, hsa-miR-620, hsa-miR-554, hsa-miR-625, hsa-miR-661, hcmv-miR-UL70-5p, hsa-miR-325, hsa-miR-453, hsa-miR-552, hsa-miR-558, hsa-miR-223, hsa-miR-302c, hsa-miR-142-5p, hsa-miR-649, hsa-miR-142-3p). Several genes were recognised targets of those microRNAs, and at least two (*LOX*, *VEGFA*) were already known to be involved in GBM tumourigenesis: *LOX*, a cofactor of *HIF-1* which helps tumour cell to adapt to hypoxia [9], and *VEGFA*, known to be upregulated in older GBMs [10]. Regarding DMGs, 184 genes were shown to be hypermethylated in the subgroup of the older patients, whereas 4 genes were in those of the younger ones. Overall, 18 of those GBM-specific DMGs were already known to have a role in carcinogenesis and included in the PubMeth database [11]. Seven of those genes were found to be specifically hypermethylated and downregulated in older GBMs as compared to younger (*MYO1B*, *PRKCB1*, *VRK2*, *FZD6*, *DLK1*, *SLC25A21*, *MSC*). Finally, the authors described significant higher rates of chromosome 10 deletions and chromosome 7 amplifications in older as compared to younger patients (which was similarly demonstrated in paediatric versus adult patients) [12], and found more mutations in the former than in the latter after whole exome sequence (WES) analysis.

In 2018, Ostrom et al. published a paper aiming to explore the relationship between the genetic risk for GBM and age-at-diagnosis [13]. Therefore, the authors performed a genome-wide association study (GWAS) of 4512 GBMs from four institutions. A significant association between single nucleotide polymorphism (SNP) variants and higher risk for GBM was confirmed for two loci located in 7p11.2 (rs723527 and rs11979158), but only in persons older than 54 years. These loci were located near the gene of the *EGFR*, which is strongly associated with risk for GBM [14]. Otherwise, a risk locus in 8q24.21(rs55705857), which was already known to be associated with lower-grade gliomas, resulted to prevail in younger patients (18–53 years). Then, the authors carried out a secondary analysis of the TCGA GBM dataset: they found that individuals between 18 and 53 years old presented *IDH*-mutant GBMs in 15/100 cases (15%), whereas older patients had a significantly lower prevalence of *IDH*-mutant tumours (2/94 (2.1%) in the 54–63-year-old group, and 1/121 (0–8%) among patients older than 64 years). 

These data reinforced the fact that *IDH*-mutant high-grade astrocytomas (formerly known as “secondary GBMs”) are separate entities from GBMs: in fact, they mostly evolve from lower-grade astrocytomas, and harbour distinct genetic and molecular features. For this reason, the 2021 WHO Classification of brain tumours recognises a new class of astrocytomas *IDH*-mutant grade 4, whereas the diagnosis of glioblastoma *IDH*-mutant is not allowed anymore [15].

Based on these considerations, Fukai et al. published a study on a selected cohort of 212 patients with *IDH*-wildtype GBM, to compare the molecular features of elderly (≥70-year-old) versus younger (≤40-year-old) patients [16]. In the elderly group, *PTEN* deletions and *CDK4* amplification prevailed significantly (46.7% versus 24.2%, *p* = 0.040, and 18.5% versus 3.0%, *p* = 0.030, respectively), and a trend for higher representation of *PDGFR* amplified/gained tumours was seen among the elderly (23.9% versus 9.1%, *p* = 0.068); likewise, mutation of *TERT* promoter was slightly prevalent among elderly patients (55.4% versus 39.3%, *p* = 0.168). Furthermore, a non-significant association between the coexistence of triple copy-number alterations (CNAs) in the *EGFR*, *CDKN2A* and *PTEN* genes was found in the older group. Conversely, *TP53* mutation was detected in the older group less frequently than in the younger group. Interestingly, *MGMT* promoter methylation was balanced in the two age groups. The authors also performed a survival analysis stratified by molecular factors: younger patients with *MGMT*p methylation and p*TERT* mutation had a significantly longer survival as compared to older patients (55.7 versus 18.7 months, *p* = 0.013, and 38.3 versus 19.6 months, *p* = 0.027, respectively), and younger patients with non-triple CNAs had a better survival than older patients (26.2 versus 19.6 months, *p* = 0.045). Notably, the benefit provided by the *MGMT*p methylation was not significant in the elderly subgroup, being overall survival 18.7 and 17.1 months among methylated and non-methylated patients, respectively.

To conclude, even if molecular characteristics of elderly GBM patients are not entirely known, there is evidence that many genetic factors negatively impact the prognosis by interacting along with other clinical features.

## 3. Prognostic Assessment 

The prognostic assessment of elderly people with GBM is of primary importance to tailor treatments to clinical status and expected outcome. Elderly patients with GBM may not tolerate aggressive regimens of radio and chemotherapy, and benefit from therapy against risk of over-treatment should be carefully balanced.

Scott et al. published a retrospective analysis of prognostic factors for GBM patients aged 70 years or older [17]. Four subgroups with different prognosis were identified, based on extent of surgery, age, and clinical status (Karnofsky performance status (KPS)) [18]: patients undergoing extended resection and age <75.5 years (median overall survival (mOS) 9.3 months), or ≥75.5 years (mOS 6.4 months); patients undergoing biopsy with KPS 70–100 (mOS 4.6 months) or KPS < 70 (mOS 2.3 months). 

However, due to the heterogeneity of the elderly population, age and KPS are not exhaustive predictors of prognosis, and other clinical factors should be considered. For example, in a large series of elderly GBM patients from the Memorial Sloan Kettering Cancer Center, the number of tumour lesions was an additional factor associated with poorer outcome, together with older age, worse KPS, and incomplete surgery [19]. Furthermore, the presence of comorbidities may negatively affect the outcome of elderly GBM patients. In a series of 129 patients older than 65 years undergoing resection, KPS < 80, COPD, motor deficit, language deficit, cognitive deficit, and tumour size larger than 4 cm had a significant negative impact on prognosis. Notably, other factors (such as age, coronary artery disease, diabetes, hypertension, atrial fibrillation, symptom duration, involvement of eloquent areas) did not seem to be of prognostic importance. Therefore, patients were divided into three groups of risk based on the presence of one or more of the aforementioned factors with effect on the outcome (group 1: 0–1 factor; group 2: 2–3 factors; group 3: 4–6 factors). Survival times significantly differed among those groups, being mOS 9.2, 5.5, and 4.4 months for groups 1, 2, and 3, respectively [20]. The impact of clinical status (according to the Eastern Cooperative Oncology Group [ECOG] score) as a prognostic factor in a multivariable analysis was also confirmed in a larger series of 339 patients > 70 years, where each ECOG level (1–4 versus 0) was increasingly associated with a worse prognosis, together with other clinical and neuroradiological factors (multifocal versus single lesions, mass effect, limited surgery, best supportive care only). However, in this series patient comorbidities did not have a prognostic importance in the multivariate analysis [21]. The Charlson Comorbidities Index (CCI) is the most common score for the assessment of patient comorbidities in the GBM setting [22]. This score includes a wide range of clinical conditions, such as age, myocardial infarction, congestive heart failure, peripheral vascular disease, cerebrovascular accidents, dementia, chronic obstructive pulmonary disease (COPD), connective tissue disease, peptic ulcer disease, liver disease, diabetes mellitus, hemiplegia, chronic kidney failure, solid tumours, blood tumours, and acquired immunodeficiency syndrome (AIDS). The sum of values given to each comorbidity determines the final CCI score, which is used to predict the expected survival of a patient. The role of CCI was investigated in few studies on elderly GBM, with unclear correlation with outcome. In a study on 146 patients with glioblastoma (including 56 patients older than 65 years), CCI < 2 did not correlate with outcome. However, only 8/56 elderly patients had a CCI > 2, which made this association uncertain [23]. Conversely, in another small study on 35 patients older than 65 years treated with radio-chemotherapy, age-adjusted CCI correlated with prognosis: patients with CCI < 3 (22/35, 62.8%) had an mOS of 22 months, while those with CCI ≥ 3 (13/35, 37.2%) had an mOS of 10 months [24]. Results from this study are peculiar for at least two reasons: first, overall survival of elderly patients with CCI < 3 was extremely long; second, the proportion of patients with CCI ≥ 3 was smaller than usually seen in similar series. Additional correlations with other clinical and molecular factors (i.e., *MGMT*p methylation), which might have explained these issues, were not investigated [24]. In another study on 233 elderly patients, those with CCI > 3 (46%) showed worse mPFS and mOS [25]. Interestingly, median age of patients included in this study (62 years) was relatively lower than that of patients included in similar series. This may suggest a higher prevalence of patients with age-independent comorbidities with worse outcome per se. Other studies have explored the correlation between comorbidities and post-operative complications on outcome of elderly GBM patients. For example, Liu et al. suggested that CCI was positively related to higher incidence of post-operative complications [26].

A standard score of baseline factors that may guide the choice of treatment in elderly GBM patients has not been assessed so far. In a study from Flanigan et al. on 161 patients, five factors were employed to predict benefit from resection and survival: age, CCI, pre-operative weakness, size of tumour, and extent of resection. One point was assigned to each of these, and the authors found that having 3 to 5 points was associated to decreased survival as compared to 0 to 2 points, and to increased survival as compared to 6 to 9 points. In addition, they found that patients with 6 to 9 score had a poor prognosis regardless of type of surgery (resection or biopsy) [27]. Likewise, in a series from Schneider et al., the combination of CCI > 2, subtotal resection, unmethylated *MGMT* promoter status, Body Mass Index (BMI) < 30, and clinical frailty (according to the modified Frailty Index (mFI)) was a predictor of worse outcome [28]. Finally, the Comprehensive Geriatric Assessment (CGA), a scale for defining frail patients which is more commonly used in other fields of oncology, has been suggested for GBM elderly patients as well. The CGA was introduced in oncology in 2000 [29]. It comprises different tests (i.e., the Cumulative Illness Rating Scale-Comorbidity and Severity Index, Activities of Daily Living, Instrumental Activities of Daily Living, the Mini Mental State Examination, and the Geriatric Depression Scale) which serve to distinguish ‘fit’, ‘vulnerable’ and ‘frail’ patient categories. In a study on 113 elderly GBM patients, ‘fit’, ‘vulnerable’ and ‘frail’ patients accounted for 5%, 30%, and 35%, respectively. On multivariate analysis, the CGA score resulted an independent predictor of survival, as ‘vulnerable’ and ‘frail’ patients had a hazard ratio of 1.5 and 2.2, respectively, compared to fit patients (*p* = 0.04). Conversely, no association between CGA and progression-free survival (PFS) was demonstrated [30]. 

A summary of the main studies investigating the prognostic factors of elderly patients with glioblastoma is reported in Table 1. 

## 4. Therapeutic Aspects

Due to aggressive tumour behaviour, clinical frailty, and dismal prognosis, treatment of elderly GBM patients represents a clinical challenge. Here, we discuss the role of different treatment strategies for this subgroup of patients.

### 4.1. The Role of Surgery

Gross total resection may be warranted, when feasible, in GBM patients. Surgery provides tumour debulking, histomolecular diagnosis, and improves survival. Whether elderly patients should avoid extended resections due to a higher risk of post-operative complications has long been debated. In a small prospective study on 30 patients older than 65 years with suspected GBM, patients were randomised to receive either stereotactic biopsy or open craniotomy. In 19 patients the diagnosis of GBM was confirmed, and they were subsequently treated with radiotherapy. Overall survival was longer in the group with extended surgery, as compared to biopsy (171 days versus 85, *p* = 0.035) [31]. In a retrospective case–control study, 40 GBM patients aged 65 years or older were matched to 40 patients stratified for age, performance status, tumour location, and adjuvant treatment. Again, patients who received large resections had a better overall survival than those receiving biopsies (5.7 versus 4.0 months, *p* = 0.020) [20]. Similarly, another retrospective trial on 142 elderly patients with newly-diagnosed GBM established the superiority of extended resection over biopsy in terms of overall survival (13.0 months versus 4.0 months, *p* < 0.001) [32]. Finally, data from randomised phase III trials NOA-08 and Nordic confirmed that surgical resection is superior to biopsy alone [33,34]. Therefore, even if data from large phase III trials are limited, there is evidence supporting the role of extended resection to improve the outcome of elderly patients with GBM. 

### 4.2. The Role of Radiotherapy

Radiotherapy is a highly effective treatment for GBM patients. Even if there is concern for reduced tolerability and higher risk for neurological deterioration after brain irradiation, safety and feasibility of radiotherapy in elderly patients have been confirmed by several studies. In a randomised trial enrolling GBM patients aged 70 years or older, radiotherapy (50 Gy distributed in 1.8 Gy fractions) in addition to supportive care was compared to best supportive care alone. Both progression-free survival and overall survival were superior after radiotherapy (14.9 versus 5.4 weeks; 29.1 versus 16.9 weeks, *p* = 0.002, respectively), while tolerability and quality of life in the two groups were similar [35]. 

Patients undergoing radiation therapy are often referred to an outpatient service that requires daily access from home: for this reason, elderly patients with frail conditions may not tolerate prolonged radiotherapy schedules. Therefore, shorter schedules have been investigated. A trial comparing 40 Gy in 15 fractions with standard 60 Gy in 30 fractions did not show any difference, overall survival being 5.6 months for the hypofractioned schedule and 5.1 months for the standard one [36]. Additionally, the Atomic Energy Agency conducted a trial comparing hypofractioned radiotherapy with 40 Gy in 15 fractions to short course radiotherapy with a 5 × 5 Gy schedule: no differences were seen between the two regimens [37]. Based on these trials, that were further corroborated by larger phase III studies [34,38], hypofractioned radiotherapy has become a standard of care for elderly GBM patients. 

### 4.3. The Role of Chemotherapy

Temozolomide has been investigated as a single adjuvant strategy in elderly patients due to the simple method of administration (being an oral agent), and low toxicity. In a phase II study on GBM patients aged 70 years or older, TMZ was administered as adjuvant monotherapy according to standard schedule (150–200 mg/m^2^/daily for 5 days every 28 days). Overall, TMZ was well tolerated, with grades 3 to 4 neutropenia and thrombocytopenia in 13 and 14% patients, respectively. In addition, improvement of performance status and quality of life were reported. As expected, efficacy of TMZ was greater among patients with *MGMT*p methylation [39]. These data confirm the efficacy of TMZ chemotherapy among elderly patients and reinforce the rationale of using it in combination with radiation (see below).

### 4.4. Radiotherapy versus Chemotherapy

Two randomised phase III trials compared the efficacy of temozolomide versus radiotherapy for elderly patients in the adjuvant setting: the Nordic trial assigned 291 GBM patients aged more than 60 years to temozolomide monotherapy (150–200 mg/m^2^/daily for 5 days every 28 days), versus standard 60 Gy/30 fractions radiotherapy, versus hypofractioned 34 Gy radiotherapy. Outcome of patients undergoing TMZ monotherapy or hypofractioned radiotherapy did not differ significantly, whereas hypofractioned radiotherapy was associated with a significantly longer overall survival than standard radiotherapy [34]. The NOA-08 trial compared adjuvant therapy with intensified TMZ (100 mg/m^2^ given on days 1–7 every other week) to standard 60 Gy radiotherapy. TMZ was not inferior to radiotherapy regarding overall survival (1-year survival being 34.4% and 37.4% for TMZ and radiotherapy groups, respectively) [33]. 

In both trials, the benefit provided by TMZ was higher in the *MGMT*p-methylated patients.

### 4.5. The Role of Post-Operative Radio-Chemotherapy

Frail elderly patients with comorbidities may not be able to tolerate intensive radio-chemotherapy regimens after surgery; moreover, subgroup analysis from the EORTC/NCIC trial failed to demonstrate a benefit in terms of survival for patients aged more than 65 years treated with combined chemoradiation over radiotherapy alone [40,41]. However, as radiotherapy and chemotherapy with temozolomide have been proven to be safe in elderly GBM, their combination, which is the standard of care in younger patients, has been proposed for elderly patients as well. In particular, the addition of temozolomide to hypofractioned radiotherapy (40 Gy in 15 fractions), as compared to hypofractioned radiotherapy alone, was investigated in the NCIC/EORTC phase III trial (NCT00482677), which enrolled 562 patients aged 65 years or older. In the experimental group, radiotherapy was associated with concomitant temozolomide (75 mg/m^2^ daily during radiotherapy),and followed by maintenance temozolomide (150–200 mg/m^2^/daily for 5 days every 28 days) for up to 12 cycles or until progression. Radio-chemotherapy resulted in higher overall survival (9.3 versus 7.6 months, HR = 0.67, *p* < 0.0001). The addition of temozolomide also resulted in a prolonged PFS of 5.3 versus 3.9 months (HR = 0.50; *p* < 0.001). The benefit from combined radio-chemotherapy was particularly significant among patients with *MGMT* promoter-methylated tumours, who displayed a median overall survival of 13.5 months compared to only 7.7 months in the radiotherapy group (HR = 0.53; *p* < 0.0001). Conversely, in patients with *MGMT* unmethylated tumours, there was a non-significant trend favouring the radio-chemotherapy group (median overall survival being 10.0 vs. 7.9 months; HR = 0.75; *p* = 0.055) [38].

To conclude, the main principles regarding current management of elderly patients with GBM may be summarised as follows: (1) based on Nordic and NCIC/EORTC (NCT00482677) phase III trials, combined hypofractioned radiotherapy and chemotherapy with TMZ followed by adjuvant TMZ may be an option in elderly patients with a good performance status, especially with *MGMT*p-methylated tumours. (2) For patients without *MGMT*p methylation, hypofractioned radiotherapy may be a reasonable option. (3) For patients with *MGMT*p methylation, but with poor clinical status or requiring an extensive field for radiotherapy, monotherapy with TMZ is an appropriate choice (Figure 1).

## 5. New Molecular Perspectives

The investigation of new pharmacological agents targeting molecular pathways that are involved in tumour development and progression is of primary importance in glioblastoma, where standard therapies are poorly effective. In the last decade, several clinical trials have explored the role of different targeted therapies. Unfortunately, the majority of those treatments have shown scarce (if not any) anti-tumour activity. In addition, results from trials investigating the effectiveness and safety of targeted therapies in cohorts of elderly glioblastoma patients are lacking. 

One of the most explored strategies is represented by the inhibition of the *EGFR* pathway. Several agents directed against the *EGFR* pathway have been investigated, such as *EGFR* inhibitors (e.g., erlotinib) [42], the anti-EGFRvIII peptide vaccine rindopepimut [43], or the antibody–drug conjugate depatuxizumab mafodotin (ABT414) [44,45,46,47], but without consistent results. Drugs targeting the *BRAF* V600E mutation (monotherapy with Raf inhibitors such as vemurafenib [48], or dual therapy with combined BRAF/MEK inhibition with trametinib and dabrafenib [45]) have been proven to be an option in tumours harbouring the mutation; however, *BRAF* mutations are a very rare finding among adult and especially elderly patients with GBM. Likewise, other potentially targetable mutations, such as *NTRK* fusions [49], *H3K27M* mutations [50], *FGFR* mutations and *FGFR3-TACC3* fusions [51], are all uncommon in glioblastoma, and virtually absent among elderly people. Of note, one phase II trial has investigated the employ of bevacizumab plus temozolomide in a cohort of 66 elderly patients (≥70 years) with newly-diagnosed GBM and poor KPS (<70), with favourable results (mPFS 15.3 weeks; mOS 23.9 weeks) and good tolerability [52]. However, several trials exploring new molecularly targeted treatments are now ongoing, and some results may be available in the near future. For example, the National Center for Tumor Diseases-Heidelberg Neuro Master Match (N2M2) (NCT03158389) is a trial currently ongoing where molecularly matched targeted therapies plus RT in patients with newly-diagnosed glioblastoma without MGMT promoter are being evaluated [53]. Similarly, the Individualized Screening Trial of Innovative Glioblastoma Therapy (INSIGhT) trial, which evaluates *EGFR*, *mTOR/DNA-PK*, and *CDK4/6* inhibitors [54], and the GBM Adaptive, Global, Innovative Learning Environment (AGILE) consortium [55], randomise patients into different arms of treatments based on the presence of biomarkers and may suggest a benefit from a specific targeted therapy. 

## 6. Conclusions

Glioblastoma of elderly patients is an aggressive condition with dismal prognosis. However, several clinical and molecular factors may help to identify patients with different response to treatment and outcome. A combined clinical and molecular score is needed to predict the outcome and suggest the best treatment strategy in this group of patients in order to increase survival, reduce toxicity, and improve quality of life.

## Figures and Tables

**Figure 1 biomedicines-10-00644-f001:**
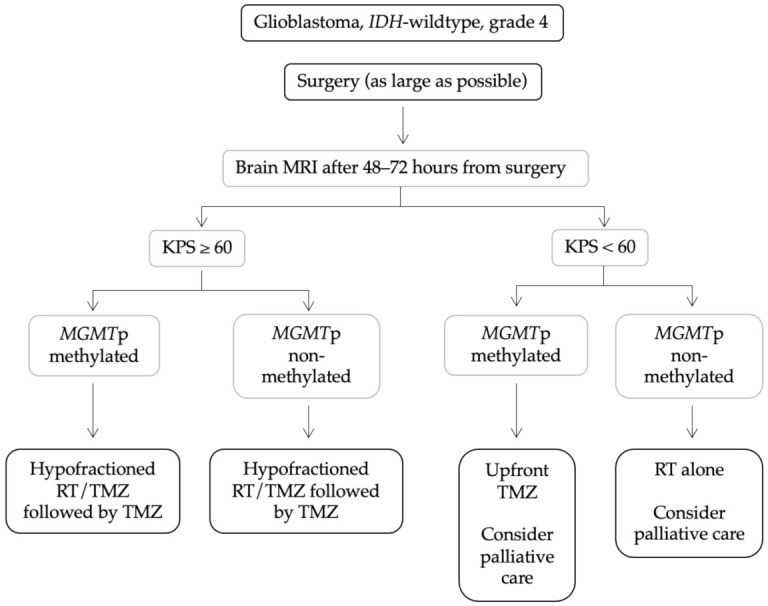
Clinical management of elderly patients with glioblastoma. Abbreviations: KPS, Karnofsky Performance Status; *MGMT*p, O-6-methylguanine-DNA methyltransferase promoter; MRI, Magnetic Resonance Imaging; RT, radiotherapy; TMZ, temozolomide.

**Table 1 biomedicines-10-00644-t001:** Main studies investigating the impact of different prognostic factors in elderly glioblastoma patients.

Author	Type of Study	No. of Patients	Age (years)	Factors Associated with Outcome	Median Overall Survival and Analysis of Survival
Scott et al., 2012 [17]	Multicentric study (MSKCC + CC; French consortium)	506	≥70	Age KPS Extent of resection	MSKCC + CC GTR/STR; Age < 75.5: 9.3 GTR/STR; Age ≥ 75.5 y: 6.4 Biopsy; KPS 70–100: 4.6 Biopsy; KPS < 70: 2.3 French consortium GTR/STR; Age < 75.5: 8.5 GTR/STR; Age ≥ 75.5 y: 7.7 Biopsy; KPS 70–100: 4.3 Biopsy; KPS < 70: 3.1
Iwamoto et al., 2009 [19]	Monoinstitutional series	394	≥65	Age KPS Multifocal tumour Extent of resection Comorbidity: no correlation	8.6 (for the whole cohort) Multivariable analysis (HR, *p* value) 70–74 yrs vs. 65–69: 1.3, *p <* 0.001 75–79 yrs vs. 65–69: 2.0, *p <* 0.001 ≥80 yrs vs. 65–69: 1.8, *p* < 0.001 KPS ≥ 70 vs. <70: 0.6, *p* < 0.001 Multifocal vs. single: 1.7, *p* < 0.001 STR vs. biopsy: 0.7, *p* = *p* < 0.001 GTR vs. biopsy: 0.5, *p* < 0.001
Chaichana et al., 2011 [20]	Monoinstitutional series	129	≥65	KPS Motor deficits Language deficits Cognitive deficits COPD Tumour size Other comorbidities (coronary artery disease; diabetes; hypertension; atrial fibrillation): no correlation	7.9 (for the whole cohort) Multivariable analysis (HR, *p* value) KPS < 80: 1.756, *p* = 0.001 Motor deficits: 3.480, *p* = 0.01 Language deficits: 2.311, *p* = 0.005 Cognitive deficits: 1.792, *p* = 0.02 COPD: 3.762, *p* = 0.01 Tumour size > 4 cm: 1.982, *p* = 0.002
Lorimer et al., 2017 [21]	Multicentric study	339	≥70	ECOG Seizures Multifocal tumour Extent of resection RT and/or TMZ vs. BSC CCI: no correlation	3.8 (for the whole cohort) Multivariable analysis (HR, *p* value) ECOG 1 vs. 0: 1.664, *p* = 0.042 ECOG 2 vs. 0: 1.780, *p* = 0.031 ECOG 3 vs. 0: 2.198, *p* = 0.008 ECOG 4 vs. 0: 2.409, *p* = 0.021 Multifocal vs. single: 3.419, *p* = 0.013 STR vs. nothing: 0.625, *p* = 0.019 GTR vs. nothing: 0.560, *p* = 0.019 RT vs. BSC: 0.588, *p* = 0.005 TMZ alone vs. BSC: 0.395, *p* = 0.004 RT/TMZ vs. BSC: 0.189, *p* < 0.001
Balducci et al., 2012 [23]	Pooled analysis from 3 phase II trials	146	≥70	Age Extent of resection RT dose CCI: no correlation	14 months (for the whole cohort) Multivariable analysis (HR, *p* value) STR vs. GTR: 1.7, *p =* 0.01 60 Gy vs. 70 Gy: 1.89, *p =* 0.01 Recursive partitioning analysis: 2.37, *p* = 0.013
Fiorentino et al., 2012 [24]	Monoinstitutional series	35	≥65	CCI	CCI < 3: 22 months CCI ≥ 3: 10 months No multivariable analysis is provided
Flanigan et al., 2018 [27]	Monoinstitutional series	161	≥65	Female sex Extent of resection Post-resection comorbidities Adjuvant TMZ vs. no TMZ	9.3 (for the whole cohort) Multivariable analysis (HR, 95% CI) Female vs. male: 0.56 (0.38–0.83) GTR vs. STR: 0.67 (0.46–0.97) No post-resection comorbidities: 0.60 (0.38–0.97) TMZ vs. no TMZ: 0.36 (0.21–0.62)
Lombardi et al., 2019 [30]	Monoinstitutional series	113	≥65	CGA KPS MGMT promoter methylation Extent of surgery: no correlation RT + TMZ: no correlation	13.2 months (for the whole cohort) Multivariable analysis (HR, *p* value) CGA (‘unfit’ vs. ‘fit’): 1.8, *p* = 0.050 *MGMT*p methylation: 0.4, *p* = 0.001 KPS ≥ 70: 0.8, *p* = 0.050

Abbreviations: CCI, Charlson Comorbidity Index; CGA, Comprehensive Geriatric Assessment; COPD, chronic obstructive pulmonary disease; ECOG, Eastern Cooperative Oncology Group; GTR, gross total resection; KPS, Karnofsky Performance Status; *MGMT*p, O-6-methylguanine-DNA methyltransferase promoter; RT, radiotherapy; STR, subtotal resection; TMZ, temozolomide.

## Data Availability

Not applicable.

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
