# Peer review of "Glioblastoma in the Elderly: Review of Molecular and Therapeutic Aspects"

_biomedicines, 2022, doi:10.3390/biomedicines10030644_

Round 1

Reviewer 1 Report

Minor comments:

The number of references is below 50, which seems scarce for a Review. 

Line 191 "may" is repeated

Line 200 The A for "Authors" should be written with lower case "a"

Table 1: the references should be listed in numerical order

Line 6 (after the table) the meaning of the sentence "Gross total resection should be warranted" seems confusing to me. May the authors wanted to mean "may be warranted"?

Author Response

Dear Reviewer 1, 

thank you so much for your kind considerations and suggestions, that enrich our paper.

  1. We added more references.
  2. Done.
  3. Done.
  4. Done.
  5. I corrected the sentence.

Best regards

Reviewer 2 Report

In this manuscript Bruno et al., have summarized very nicely the clinical and molecular characteristics, current therapeutic options, and prognostic factors of elderly patients with GBM.

Minor comment:

Authors are suggested to include a illustration to highlight the therapeutic options (in a figure fomat)

Author Response

Dear Reviewer 2, 

thank you so much for your consideration. 

We added a figure with a flow-chart of the therapeutic options of elderly glioblastoma patients.

Best regards 

Reviewer 3 Report

this review article is well written and organized. I think that this paper would be of interest for scientists working in this area.
I think that a molecular perspective on new intriguing drug targets is useful to enrich the quality. please add a section. 

drugs in clinical trials must be added.

Author Response

Dear Reviewer 3, 

thank you so much for your kind considerations and suggestions, that enrich our paper.

We added a brief section on the topic of new molecular perspectives at the end of the review (see new version of the manuscript).

Best regards

This manuscript is a resubmission of an earlier submission. The following is a list of the peer review reports and author responses from that submission.